# Multicriteria Group Decision Making Based on Intuitionistic Normal Cloud and Cloud Distance Entropy

**DOI:** 10.3390/e24101396

**Published:** 2022-10-01

**Authors:** Wei Li, Yingqi Lu, Chengli Fan, Yong Heng, Xiaowen Zhu

**Affiliations:** 1School of Air Defense and Anti-Missile, Airforce Engineering University, Xi’an 710051, China; 2Beijing Institute of Electronic System Engineering, Beijing 100854, China

**Keywords:** backward cloud generation algorithm, cloud distance entropy, intuitionistic normal cloud, VIKOR, multicriteria group decision making

## Abstract

The uncertainty of information is an important issue that must be faced when dealing with decision-making problems. Randomness and fuzziness are the two most common types of uncertainty. In this paper, we propose a multicriteria group decision-making method based on intuitionistic normal cloud and cloud distance entropy. First, the backward cloud generation algorithm for intuitionistic normal clouds is designed to transform the intuitionistic fuzzy decision information given by all experts into an intuitionistic normal cloud matrix to avoid the loss and distortion of information. Second, the distance measurement of the cloud model is introduced into the information entropy theory, and the concept of cloud distance entropy is proposed. Then, the distance measurement for intuitionistic normal clouds based on numerical features is defined and its properties are discussed, based on which the criterion weight determination method under intuitionistic normal cloud information is proposed. In addition, the VIKOR method, which integrates group utility and individual regret, is extended to the intuitionistic normal cloud environment, and thus the ranking results of the alternatives are obtained. Finally, the effectiveness and practicality of the proposed method are demonstrated by two numerical examples.

## 1. Introduction

Multicriteria group decision making (MCGDM) is an important research topic in modern decision science, and its theory and methods have been widely used in economic and social fields [1]. Many scholars have conducted in-depth studies on group decision-making problems in different decision-making environments and proposed corresponding MCGDM techniques [2,3,4,5]. In realistic MCGDM, the scale of the decision-making group grows larger as the decision environment becomes more complex [6]. Decision information has a higher degree of uncertainty because decision makers can recognize the limitations of their abilities and experience [7]. To describe the vagueness and uncertainty in decision making, mathematicians, led by Zadeh, created fuzzy set theory [8], which provides an effective tool for decision theory. In 1986, Atanassov [9] proposed the concept of the intuitionistic fuzzy set (IFS) (the vague set proposed by Gau and Buehrer [10] is actually IFS [11]). IFS considers membership degree, nonmembership degree, and hesitation degree at the same time, and is more flexible and practical than traditional fuzzy sets in dealing with uncertainty. Reference [12] proposed an intuitionistic fuzzy group decision-making method based on evidence theory. Reference [13] proposed a dynamic intuitionistic fuzzy group decision-making method and applied it to air defense threat assessment. Reference [14] extended the TOPSIS method to the intuitionistic fuzzy environment and proposed an intuitionistic fuzzy multiattribute group decision-making method based on TOPSIS. Reference [15] proposed a risk ranking method based on intuitionistic fuzzy multiattribute group decision making. In addition, as an extension of intuitionistic fuzzy sets, the linguistic intuitionistic fuzzy set has also attracted extensive attention and research [16,17]. In actual decision making, the evaluation environment not only has a certain degree of ambiguity but also contains a large amount of random information. Although the existing methods have certain advantages for the processing of fuzzy information, they only consider the fuzziness of things and ignore the randomness of things. Especially as the size of the decision-making group increases, the role of randomness in the group decision-making process becomes more prominent [18].

The information aggregation of multiple experts is another key problem to be solved in MCGDM [19]. The cloud model is an uncertainty transformation model of qualitative concepts and quantitative values proposed by Li [20] based on probability theory and fuzzy mathematics, and its feature of taking into account fuzziness and randomness provides new ideas and methods for the aggregation of decision information. The uncertain evaluation information given by multiple decision makers is regarded as the set of partial cloud drops in the cloud model, and the numerical features of the corresponding cloud model are obtained through the backward cloud generation algorithm. The most basic and common cloud model is the normal cloud (NC) model, which is also due to the universality and breadth of the normal distribution [21]. Reference [22] constructed a comprehensive evaluation cloud according to the backward cloud generation algorithm and weighted integration technology, and then proposed a statistical information quality evaluation method according to the similarity between the comprehensive cloud and the evaluation grade cloud model. Reference [23] proposed a collapse risk assessment method for highways based on an improved backward cloud generation algorithm considering internal and trigger factors. Reference [24] proposed a backward cloud generation algorithm of a neutrosophic normal cloud and constructed a multicriteria group decision-making approach to single-value neutrosophic environments. Reference [25] combined AHP, Delphi, and cloud models to propose a cloud clustering-based group decision method, in which a backward cloud generator is used to calculate the digital characteristics of the sample from the experts’ scoring. However, the current backward cloud generation algorithms can only deal with data such as exact numbers, neutrosophic numbers, and interval numbers [26], and there is a lack of research on backward cloud generation algorithms for intuitionistic fuzzy environments. In addition, similar to fuzzy sets, the normal cloud model has difficulty in dealing with the nonmembership and hesitation of fuzzy concepts. In particular, when the element values are equal to the expected values, the membership degree is the exact value 1. To this end, reference [27] combined intuitionistic fuzzy theory with cloud theory and proposed the intuitionistic normal cloud (INC) model, which achieves an accurate description of decision information uncertainty through five numerical features: expectation, membership degree, nonmembership degree, entropy, and hyper entropy. In light of the above analysis, we designed the backward cloud generation algorithm for intuitionistic normal clouds to realize the aggregation of multiple experts’ decision information in group decision making in the intuitionistic fuzzy environment.

In the research of MCGDM methods, the determination of criteria weight is a key factor affecting the objectivity of decision making. Information entropy, as a tool to measure the amount of information, can reduce the influence of subjective factors in the decision-making process, and has been widely used in weight determination [28,29,30]. Existing weight determination methods based on information entropy can describe the quality of information, but they do not reflect the preference information and group consistency well and are difficult to adapt to the decision-making problem in the cloud environment. Therefore, we combine the cloud distance measurement and information entropy and propose a method to calculate the criterion weight based on cloud distance entropy. Achieving the objective measurement of the amount of information for decision making in the cloud environment through entropy effectively improves the accuracy and effectiveness of MCGDM.

In MCGDM problems, the ranking of alternatives is also an important part of the decision-making process. Vlsekriterijumska Optimizacija I Kompromisno Resenje (VIKOR) is a compromise ranking method that integrates group utility and individual regret, proposed by Opricovic in 1998 [31]. VIKOR can adequately balance the relationship between groups and individuals and has been widely used in multicriteria decision-making problems [32,33,34]. However, in previous research and applications, VIKOR is often combined with fuzzy set theory to achieve the evaluation and ranking of solutions, and it is difficult to deal with decision-making problems in which the decision matrix is the cloud model. In view of this, we propose the VIKOR ranking method in the intuitionistic normal cloud environment to solve the multicriteria group decision-making problem in the intuitionistic normal cloud environment more effectively.

In summary, the motivation and contribution of the work in this paper are as follows. To fully consider the fuzziness and randomness of multicriteria group decision making in the intuitionistic fuzzy environment and overcome the deficiency of existing backward cloud generation algorithms that have difficulty handling the intuitionistic fuzzy numbers, we designed the backward cloud generation algorithm of intuitionistic normal cloud for aggregating expert evaluation information, which can avoid the loss and distortion of information. To overcome the shortcoming that information entropy ignores group consistency when measuring information quality in the cloud environment, we propose the concept of cloud distance entropy, which extends the flexibility of the traditional entropy method in decision making. In addition, we propose the VIKOR method in the intuitionistic normal cloud environment, which provides a new idea to solve the multicriteria group decision-making problem where the decision information is in the form of cloud representation.

The rest of the paper is structured as follows. In Section 2, we briefly introduce the basic intuitionistic fuzzy set theory, cloud model theory, and intuitionistic normal cloud model theory. Section 3 first designs the backward cloud generation algorithm for intuitionistic normal clouds, and then gives the definition of cloud distance entropy and the distance measurement for intuitionistic normal clouds. On this basis, the criterion weight calculation method and the VIKOR multicriteria group decision method in the intuitionistic normal cloud environment are proposed. In Section 4, the validity of the proposed method is verified by two illustrative numerical examples. Finally, the conclusions of this paper are given in Section 5.

## 2. Preliminaries

### 2.1. Intuitionistic Fuzzy Set

Intuitionistic fuzzy set, as a generalization and extension of fuzzy sets, describes the essential properties of things in more detail through nonmembership functions and hesitation functions. It is defined as follows:

**Definition** **1**([35])**.**
*Let*
X
*be a universe of discourse, The IFS* A *in* X
*can be defined as follows:*
(1)$$A=\left\{\langle x,u_{A}(x),v_{A}(x)\rangle |x\in X\right\}$$
*where*
$u_{A}(x)\in [0,1]$
*and*
$v_{A}(x)\in [0,1]$
*are membership function and nonmembership function of element*
x
*to set*
A*, respectively, under the condition*
$0\leq u_{A}(x)+v_{A}(x)\leq 1$*, for all x∈X.*

### 2.2. Cloud Model

The cloud model is a mathematical model that can relate qualitative linguistic values to quantitative values through uncertain relationships. The model can better describe the relationship between the ambiguity and randomness of things. It is defined as follows:

**Definition** **2**([36])**.**
*Let* U *be a quantitative universe represented by exact numerical values and* C *be a qualitative concept on* U*. If the quantitative value* x∈U *is a random realization of the qualitative concept* C*, the degree of certainty* u(x) *of* x *on* C *is a random number that tends to be stable, where* u:U→[0,1],∀x∈U,x→u(x)*, then the distribution of* x *on the universe* U *is called a cloud, and each* x *is called a cloud droplet.*

The numerical characters of the cloud are represented by three numbers [37]: expectation Ex, entropy En, and hyperentropy He. Among them, expectation Ex is the central value of the conceptual domain of qualitative language; entropy En is a measure of qualitative conceptual ambiguity; hyperentropy He reflects the degree of dispersion of cloud drops and the random change of membership. It can be seen that the three numerical eigenvalues of the cloud model integrate ambiguity and randomness in qualitative and quantitative transformation.

### 2.3. Intuitionistic Normal Cloud Model

The intuitionistic normal cloud overcomes the deficiency that the normal cloud model cannot reflect the degree of nonmembership and hesitation of fuzzy concepts by combining the intuitionistic fuzzy theory and cloud theory. Its basic definition is as follows:

**Definition** **3**([27])**.**
X *is a given domain of discourse, and* T *is a qualitative concept related to the domain of discourse.* Y=(<Ex,[u,v]>,En,He) *is called the intuitionistic normal cloud defined on* X *corresponding to the concept* T*, where expectation* Ex*, entropy* En*, and hyperentropy* He *have the same meanings as the normal cloud model,* u *and* v *are the membership degree and nonmembership degree of* x=Ex*, respectively. Furthermore, when* u=1 *and* v=0*, the intuitionistic normal cloud model degenerates into the normal cloud model.*

Figure 1 shows the comparison of normal cloud $Y_{1}=(3,0.8,0.05)$ and intuitionistic normal cloud $Y_{2}=(<7,[0.7,0.1]>0.8,0.05)$, each containing 1000 cloud droplets. Obviously, $Y_{2}$ has lower conceptual membership and higher uncertainty than $Y_{1}$, and the thickness of the cloud also increases, which reflects the characteristics of intuitionistic fuzzy theory.

**Definition** **4**([27])**.**
*Let*
$Y_{i}=(<Ex_{i},[u_{i},v_{i}]>,En_{i},He_{i})(i=1,2,\cdots ,n)$
*be a set of INCs on the same universe of discourse, and define the intuitionistic fuzzy cloud weighted arithmetic averaging (INCWAA) operator as follows:*
(2)$$INCWAA_{\omega }(Y_{1},Y_{2},\cdots ,Y_{n})=(<\sum_{i=1}^{n}\omega_{i}Ex_{i},[\frac{\sum_{i=1}^{n}\omega_{i}u_{i}Ex_{i}}{\sum_{i=1}^{n}\omega_{i}Ex_{i}},\frac{\sum_{i=1}^{n}\omega_{i}v_{i}Ex_{i}}{\sum_{i=1}^{n}\omega_{i}Ex_{i}}]>,\sqrt{\sum_{i=1}^{n}\omega_{i}^{2}En_{i}{}^{2}},\sqrt{\sum_{i=1}^{n}\omega_{i}^{2}He_{i}{}^{2}})$$
*where*
$\omega_{i}\in [0,1](i=1,2,\cdots ,n),\sum_{i=1}^{n}\omega_{i}=1$
*is the weighted vector of $Y_{i}(i=1,2,\cdots ,n)$.*

## 3. MCGDM Model Based on Intuitionistic Normal Cloud and Cloud Distance Entropy

### 3.1. Backward Cloud Generation Algorithm for INCs

The backward cloud generation algorithm is able to aggregate extended forms of data describing qualitative concepts into a dense form describing the same concept. In the context of group decision making, the backward cloud generation algorithm can aggregate evaluation information provided by different experts into a single cloud, which takes into account both tendency and spread of the assessments. We designed a backward cloud generation algorithm for INCs, the basic idea of which is to transform a set of intuitionistic fuzzy numbers into a corresponding intuitionistic normal cloud. The backward cloud generator algorithm can be implemented by the following steps.

Step1: Calculate the sample mean $\bar{X}=\frac{1}{N}\sum_{i=1}^{N}x_{i}$ and the sample variance $S^{2}=\frac{1}{N-1}\sum_{i=1}^{N}{\left(x_{i}-\bar{X}\right)}^{2}$;

Step2: Estimate the value of Ex, u, v. The expectation Ex is the value in the cloud model that best reflects the qualitative concept, which can be estimated with the sample mean. Since u and 1−v respectively represent the lower and upper limits of membership, the smaller the distance between x and Ex, the higher the reliability of the corresponding membership of the sample, which is more important when restoring the numerical features of the intuitionistic normal cloud.
(3)$$Ex_{e}=\bar{X}$$
(4)$$u_{e}=\frac{\sum_{i=1}^{N}\frac{u_{i}}{\left|x_{i}-Ex_{e}\right|}}{\sum_{i=1}^{N}\frac{1}{\left|x_{i}-Ex_{e}\right|}}$$
(5)$$v_{e}=1-\frac{\sum_{i=1}^{N}\frac{1-v_{i}}{\left|x_{i}-Ex_{e}\right|}}{\sum_{i=1}^{N}\frac{1}{\left|x_{i}-Ex_{e}\right|}}$$

Step3: Estimate the value of En, He.
(6)$$En_{e}=\sqrt{\frac{\pi }{2}}\times \frac{1}{N}\sum_{i=1}^{N}\left|x_{i}-Ex_{e}\right|$$
(7)$$He_{e}=\sqrt{\left|S^{2}-En_{e}{}^{2}\right|}$$

Output: The estimated value $\left(<Ex_{e},[u_{e},v_{e}]>,En_{e},He_{e}\right)$ of (<Ex,[u,v]>,En,He).

### 3.2. Cloud Distance Entropy

In information theory, information entropy is an effective way to measure the degree of information uncertainty and system disorder [38], which can effectively reduce the influence of human subjective factors and is mostly used as a tool for determining weight in the field of decision making. The smaller the entropy value, the greater the degree of variation of the criterion, the more informative it is, and the greater its role in decision making, and thus the greater its weight. In addition, related information entropy [39], logical entropy [40], and other extended concepts of information entropy have also appeared in different decision-making environments.

According to competitive equilibrium theory in economics [41], the market makes the system converge to a steady state by reducing the price gap between commodities. Similarly, in the process of decision information fusion in a cloud environment, each information unit is similar to a molecule in the system. The clouds in the fusion set reduce the distance between each other by mutual attraction and increase the stability of the system by reducing the distance, thus achieving a stable state with minimum entropy and maximum information. Based on the above ideas, we introduce the cloud distance measure into the information entropy theory and propose the concept of cloud distance entropy.

**Definition** **5.***Let there be* n *sequences*$\left(s_{1},s_{2},\cdots ,s_{n}\right)$*in the fusion set, and each sequence contains*m*clouds*$\left(Y_{1},Y_{2},\cdots ,Y_{m}\right)$*.*$Y_{j}^{*}$*is the optimal cloud of sequence*$s_{j}$*.*$d(Y_{ij},Y_{j}^{*})$*is the distance between the cloud*$Y_{ij}$*and the optimal cloud*$Y_{j}^{*}$*. Take the ratio of*$d(Y_{ij},Y_{j}^{*})$*to*$\sum_{i=1}^{n}d(Y_{ij},Y_{j}^{*})$*as the probability of*$Y_{ij}$*occurs. Then the cloud distance entropy of sequence*$s_{j}$*is defined as:*(8)$$E_{j}=-\frac{1}{\ln m}\sum_{i=1}^{m}(\frac{d(Y_{ij},Y_{j}^{*})}{\sum_{i=1}^{m}d(Y_{ij},Y_{j}^{*})}\cdot \ln\frac{d(Y_{ij},Y_{j}^{*})}{\sum_{i=1}^{m}d(Y_{ij},Y_{j}^{*})}),j=1,2,\cdots ,n$$

In particular, when $Y_{1j}=Y_{2j}=\cdots =Y_{mj}=Y_{j}^{*}$, $\lim d(Y_{ij},Y_{j}^{*})/\sum_{i=1}^{m}d(Y_{ij},Y_{j}^{*})=1/m$. Based on the cloud distance entropy, the entropy weight method can be used to calculate the weight of each sequence. The entropy weight method is a method to determine the weight of indicators based on the size of the information carrying capacity of each indicator; the smaller the entropy, the greater the amount of information and the greater the weight. The objective weight of sequence $s_{j}$ is as follows.
(9)$$w_{j}=\frac{1-E_{j}}{n-\sum_{j=1}^{n}E_{j}}$$

The entropy weight method based on cloud distance entropy can comprehensively reflect the preference and quality of objective decision information, and provide the guarantee of accuracy and objectivity for the determination of criterion weights in the cloud environment.

### 3.3. Distance Measurement for INCs

Distance is an effective way to measure the difference between cloud models and an important technique to apply cloud models to solve practical problems [42,43]. In this paper, a distance measurement for intuitionistic normal clouds is proposed for the practical characteristics of intuitionistic normal clouds based on a comprehensive consideration of the importance of numerical features as follows.

**Definition** **6.***Let* $Y_{1}=(<Ex_{1},[u_{1},v_{1}]>,En_{1},He_{1})$ *and* $Y_{2}=(<Ex_{2},[u_{2},v_{2}]>,En_{2},He_{2})$ *be two INCs, then the distance measurement is defined as follows:*(10)$$d(Y_{1},Y_{2})=\sqrt{{\left(\frac{1-v_{1}}{2-u_{1}-v_{1}}Ex_{1}-\frac{1-v_{2}}{2-u_{2}-v_{2}}Ex_{2}\right)}^{2}+{\left(En_{1}-En_{2}\right)}^{2}+{\left(\sqrt{En_{1}^{2}+He_{1}^{2}}-\sqrt{En_{2}^{2}+He_{2}^{2}}+En_{2}-En_{1}\right)}^{2}}$$

In particular, when $En_{1}=En_{2}=He_{1}=He_{2}=0$, the distance between two intuitionistic normal clouds is:(11)$$d(Y_{1},Y_{2})=\left|\frac{1-v_{1}}{2-u_{1}-v_{1}}Ex_{1}-\frac{1-v_{2}}{2-u_{2}-v_{2}}Ex_{2}\right|$$

Furthermore, when $En_{1}=En_{2}=He_{1}=He_{2}=0$ and $u_{1}=u_{2}=1$, $v_{1}=v_{2}=0$, then the distance further degenerates into distance between two real numbers:(12)$$d(Y_{1},Y_{2})=\left|Ex_{1}-Ex_{2}\right|$$

**Theorem** **1.***Let* $Y_{1}=(<Ex_{1},[u_{1},v_{1}]>,En_{1},He_{1})$ *and* $Y_{2}=(<Ex_{2},[u_{2},v_{2}]>,En_{2},He_{2})$ *be two INCs;* $d(Y_{1},Y_{2})$ *is the distance between two clouds, then* $d(Y_{1},Y_{2})$ *satisfies the following properties:**(1)* $d(Y_{1},Y_{2})\geq 0$*;**(2)* $d(Y_{1},Y_{2})=d(Y_{2},Y_{1})$*;**(3)* *If and only if $Y_{1}=Y_{2}$, $d(Y_{1},Y_{2})=0$;**(4)* *If*$Y_{3}$*is an arbitrary INC, then*$d(Y_{1},Y_{2})+d(Y_{2},Y_{3})\geq d(Y_{1},Y_{3})$*.*

In order to verify the validity of the proposed distance measure, the proof process of Theorem 1 is given as follows.

**Proof of Theorem** **1.**(1) According to Definition 6 we can obtain


$$d(Y_{1},Y_{2})=\sqrt{{\left(\frac{1-v_{1}}{2-u_{1}-v_{1}}Ex_{1}-\frac{1-v_{2}}{2-u_{2}-v_{2}}Ex_{2}\right)}^{2}+{\left(En_{1}-En_{2}\right)}^{2}+{\left(\sqrt{En_{1}^{2}+He_{1}^{2}}-\sqrt{En_{2}^{2}+He_{2}^{2}}+En_{2}-En_{1}\right)}^{2}}\geq 0$$


(2) According to Definition 6 we can obtain
$$\begin{array}{l} d(Y_{1},Y_{2})\\ =\sqrt{{\left(\frac{1-v_{1}}{2-u_{1}-v_{1}}Ex_{1}-\frac{1-v_{2}}{2-u_{2}-v_{2}}Ex_{2}\right)}^{2}+{\left(En_{1}-En_{2}\right)}^{2}+{\left(\sqrt{En_{1}^{2}+He_{1}^{2}}-\sqrt{En_{2}^{2}+He_{2}^{2}}+En_{2}-En_{1}\right)}^{2}}\\ =\sqrt{{\left(\frac{1-v_{2}}{2-u_{2}-v_{2}}Ex_{2}-\frac{1-v_{1}}{2-u_{1}-v_{1}}Ex_{1}\right)}^{2}+{\left(En_{2}-En_{1}\right)}^{2}+{\left(\sqrt{En_{2}^{2}+He_{2}^{2}}-\sqrt{En_{1}^{2}+He_{1}^{2}}+En_{1}-En_{2}\right)}^{2}}\\ =d(Y_{2},Y_{1}) \end{array}$$

(3) When $Y_{1}=Y_{2}$, we can obtain
$$d(Y_{1},Y_{2})=\sqrt{{\left(\frac{1-v_{1}}{2-u_{1}-v_{1}}Ex_{1}-\frac{1-v_{2}}{2-u_{2}-v_{2}}Ex_{2}\right)}^{2}+{\left(En_{1}-En_{2}\right)}^{2}+{\left(\sqrt{En_{1}^{2}+He_{1}^{2}}-\sqrt{En_{2}^{2}+He_{2}^{2}}+En_{2}-En_{1}\right)}^{2}}=0$$

When $Y_{1}\neq Y_{2}$, There are $\frac{1-v_{1}}{2-u_{1}-v_{1}}Ex_{1}-\frac{1-v_{2}}{2-u_{2}-v_{2}}Ex_{2}$, $En_{1}-En_{2}$, $\sqrt{En_{1}^{2}+He_{1}^{2}}-\sqrt{En_{2}^{2}+He_{2}^{2}}+En_{2}-En_{1}$ not equal to 0 at the same time, so we can obtain
$$d(Y_{1},Y_{2})=\sqrt{{\left(\frac{1-v_{1}}{2-u_{1}-v_{1}}Ex_{1}-\frac{1-v_{2}}{2-u_{2}-v_{2}}Ex_{2}\right)}^{2}+{\left(En_{1}-En_{2}\right)}^{2}+{\left(\sqrt{En_{1}^{2}+He_{1}^{2}}-\sqrt{En_{2}^{2}+He_{2}^{2}}+En_{2}-En_{1}\right)}^{2}}\neq 0$$

(4) According to Definition 6 and Cauchy–Schwarz inequality, we can obtain
$$\begin{array}{l} d(Y_{1},Y_{2})+d(Y_{2},Y_{3})\\ =\sqrt{{\left(\frac{1-v_{1}}{2-u_{1}-v_{1}}Ex_{1}-\frac{1-v_{2}}{2-u_{2}-v_{2}}Ex_{2}\right)}^{2}+{\left(En_{1}-En_{2}\right)}^{2}+{\left(\sqrt{En_{1}^{2}+He_{1}^{2}}-\sqrt{En_{2}^{2}+He_{2}^{2}}+En_{2}-En_{1}\right)}^{2}}\\ +\sqrt{{\left(\frac{1-v_{2}}{2-u_{2}-v_{2}}Ex_{2}-\frac{1-v_{3}}{2-u_{3}-v_{3}}Ex_{3}\right)}^{2}+{\left(En_{2}-En_{3}\right)}^{2}+{\left(\sqrt{En_{2}^{2}+He_{2}^{2}}-\sqrt{En_{3}^{2}+He_{3}^{2}}+En_{3}-En_{2}\right)}^{2}}\\ =\left|\alpha \right|+\left|\beta \right|\\ =\sqrt{{\left|\alpha \right|}^{2}+2\left|\alpha \right|\left|\beta \right|+{\left|\beta \right|}^{2}}\\ \geq \sqrt{{\left|\alpha \right|}^{2}+2(\alpha ,\beta )+{\left|\beta \right|}^{2}}\\ =\sqrt{\left(\alpha +\beta ,\alpha +\beta \right)}\\ =\left|\alpha +\beta \right|\\ =\sqrt{{\left(\frac{1-v_{1}}{2-u_{1}-v_{1}}Ex_{1}-\frac{1-v_{3}}{2-u_{3}-v_{3}}Ex_{3}\right)}^{2}+{\left(En_{1}-En_{3}\right)}^{2}+{\left(\sqrt{En_{1}^{2}+He_{1}^{2}}-\sqrt{En_{3}^{2}+He_{3}^{2}}+En_{3}-En_{1}\right)}^{2}}\\ =d(Y_{1},Y_{3}) \end{array}$$
where $\alpha =((\frac{1-v_{1}}{2-u_{1}-v_{1}}Ex_{1}-\frac{1-v_{2}}{2-u_{2}-v_{2}}Ex_{2},En_{1}-En_{2},\sqrt{En_{1}^{2}+He_{1}^{2}}-\sqrt{En_{2}^{2}+He_{2}^{2}}+En_{2}-En_{1})$, $\beta =((\frac{1-v_{2}}{2-u_{2}-v_{2}}Ex_{2}-\frac{1-v_{3}}{2-u_{3}-v_{3}}Ex_{3},En_{2}-En_{3},\sqrt{En_{2}^{2}+He_{2}^{2}}-\sqrt{En_{3}^{2}+He_{3}^{2}}+En_{3}-En_{2})$, (α,β) is the scalar product of vector α and vector β and |•| is the norm of vector. □

### 3.4. VIKOR Method in Intuitionistic Normal Cloud Environment

In this paper, the proposed cloud distance measurement is used to calculate the distance between each alternative and the positive and negative ideal solutions, and then the VIKOR method in the intuitionistic normal cloud environment is proposed.

For an intuitionistic fuzzy MCGDM problem, let $A=\left\{A_{1},A_{2},\cdots ,A_{m}\right\}$ be the set of m alternatives, $D=\left\{D_{1},D_{2},\cdots ,D_{p}\right\}$ be the set of p decision makers, $C=\left\{c_{1},c_{2},\cdots ,c_{n}\right\}$ be the set of n criteria. $z_{ijk}=(x_{ijk},[u_{ijk},v_{ijk}])$ is the evaluation information of the alternative $A_{i}$ given by expert $D_{k}$ under the criterion $c_{j}$, where $x_{ijk}$ represents the evaluation value given by the expert, $u_{ijk}$ and $v_{ijk}$ represent the membership degree and nonmembership degree of the evaluation value, respectively. In decision making, the degree of membership is usually used to indicate the degree of certainty of the decision maker to a certain judgment. Thus, $u_{ijk}$ and $v_{ijk}$ denote the degree of certainty and uncertainty of the expert about the given evaluation information, respectively. The decision-making steps based on the proposed method are as follows:

Step1: According to the backward cloud generation algorithm for intuitionistic normal clouds, all expert evaluation information is transformed into the intuitionistic normal cloud matrix.
(13)$$Y=\left[\begin{array}{cccc} Y_{11}(\langle Ex_{11},[u_{11},v_{11}]\rangle ,En_{11},He_{11}) & Y_{12}(\langle Ex_{12},[u_{12},v_{12}]\rangle ,En_{12},He_{12}) & \cdots & Y_{1n}(\langle Ex_{1n},[u_{1n},v_{1n}]\rangle ,En_{1n},He_{1n})\\ Y_{21}(\langle Ex_{21},[u_{21},v_{21}]\rangle ,En_{21},He_{21}) & Y_{22}(\langle Ex_{22},[u_{22},v_{22}]\rangle ,En_{22},He_{22}) & \cdots & Y_{2n}(\langle Ex_{2n},[u_{2n},v_{2n}]\rangle ,En_{2n},He_{2n})\\ \vdots & \vdots & \ddots & \vdots \\ Y_{m1}(\langle Ex_{m1},[u_{m1},v_{m1}]\rangle ,En_{m1},He_{m1}) & Y_{m2}(\langle Ex_{m2},[u_{m2},v_{m2}]\rangle ,En_{m2},He_{m2}) & \cdots & Y_{mn}(\langle Ex_{mn},[u_{mn},v_{mn}]\rangle ,En_{mn},He_{mn}) \end{array}\right]$$

Step2: Determine the positive ideal solution $Y^{+}=(Y_{1}^{+},Y_{2}^{+},\cdots ,Y_{n}^{+})$ and negative ideal solution $Y^{-}=(Y_{1}^{-},Y_{2}^{-},\cdots ,Y_{n}^{-})$ of the decision cloud matrix.
(14)$$Y_{j}^{+}=(\langle Ex_{j}^{+},[u_{j}^{+},v_{j}^{+}]\rangle ,En_{j}^{+},He_{j}^{+})=(\langle \underset{i}{\max}Ex_{ij},[\underset{i}{\max}u_{ij},\underset{i}{\min}v_{ij}]\rangle ,\underset{i}{\min}En_{ij},\underset{i}{\min}He_{ij})$$
(15)$$Y_{j}^{-}=(\langle Ex_{j}^{-},[u_{j}^{-},v_{j}^{-}]\rangle ,En_{j}^{-},He_{j}^{-})=(\langle \underset{i}{\min}Ex_{ij},[\underset{i}{\min}u_{ij},\underset{i}{\max}v_{ij}]\rangle ,\underset{i}{\max}En_{ij},\underset{i}{\max}He_{ij})$$

Step3: The distance between the alternative cloud $Y_{ij}$ and the optimal cloud $Y_{j}^{*}$ (i.e., positive ideal solution $Y_{j}^{+}$) is obtained by the distance measurement given by Equation (10).
(16)$$d(Y_{ij},Y_{j}^{*})=\sqrt{{\left(\frac{1-v_{ij}}{2-u_{ij}-v_{ij}}Ex_{1}-\frac{1-v_{j}^{+}}{2-u_{j}^{+}-v_{j}^{+}}Ex_{j}^{+}\right)}^{2}+{\left(En_{1}-En_{j}^{+}\right)}^{2}+{\left(\sqrt{En_{1}^{2}+He_{1}^{2}}-\sqrt{{\left(En_{j}^{+}\right)}^{2}+{\left(He_{j}^{+}\right)}^{2}}+En_{j}^{+}-En_{1}\right)}^{2}}$$

Step4: Calculate the cloud distance entropy $E_{j}$ of criterion $c_{j}$.
(17)$$E_{j}=-\frac{1}{\ln m}\sum_{i=1}^{m}(\frac{d(Y_{ij},Y_{j}^{*})}{\sum_{i=1}^{m}d(Y_{ij},Y_{j}^{*})}\cdot \ln\frac{d(Y_{ij},Y_{j}^{*})}{\sum_{i=1}^{m}d(Y_{ij},Y_{j}^{*})}),j=1,2,\cdots ,n$$

Step5: Calculate the weight $w_{j}$ of criterion $c_{j}$.
(18)$$w_{j}=\frac{1-E_{j}}{n-\sum_{j=1}^{n}E_{j}}$$

Step6: Calculate the group utility value $S_{i}$, the individual regret value $R_{i}$, and the compromise value $Q_{i}$ through the proposed distance measurement for intuitionistic normal clouds.
(19)$$S_{i}=\sum_{j=1}^{n}w_{j}\frac{d(Y_{ij},Y_{j}^{+})}{d(Y_{j}^{-},Y_{j}^{+})}$$
(20)$$R_{i}=\underset{j}{\max}w_{j}\frac{d(Y_{ij},Y_{j}^{+})}{d(Y_{j}^{-},Y_{j}^{+})}$$
(21)$$Q_{i}=u\frac{S_{i}-S^{-}}{S^{+}-S^{-}}+(1-u)\frac{R_{i}-R^{-}}{R^{+}-R^{-}}$$
where $S^{+}=\underset{i}{\min}S_{i}$, $S^{-}=\underset{i}{\max}S_{i}$, $R^{+}=\underset{i}{\min}R_{i}$, $R^{-}=\underset{i}{\max}R_{i}$, and u∈[0,1] is the compromise coefficient.

Step7: $A^{\left(1\right)},A^{\left(2\right)},\cdots ,A^{\left(m\right)}$ is the result of ranking $Q_{i}$ in ascending order. If $A^{\left(1\right)}$ is the optimal alternative and satisfies the following conditions: 1—$Q(A^{\left(2\right)})-Q(A^{\left(1\right)})\geq 1/(m-1)$; 2—according to the ranking results of $S_{i}$ and $R_{i}$, the alternative $A^{\left(1\right)}$ is still the optimal alternative, then the alternative $A^{\left(1\right)}$ is the most stable optimal compromise alternative.

If one of the above two conditions is not satisfied, multiple compromise alternatives are obtained:

(1) If condition 2 is not satisfied, then both $A^{\left(1\right)}$ and $A^{\left(2\right)}$ are compromise alternatives.

(2) If condition 1 is not satisfied, X is the maximum value that satisfies the condition $Q(A^{\left(X\right)})-Q(A^{\left(1\right)})\geq 1/(m-1)$, then $A^{\left(1\right)},A^{\left(2\right)},\cdots ,A^{\left(X\right)}$ are compromise alternatives.

The flow of the proposed method is shown in Figure 2.

## 4. Numerical Examples and Comparative Analysis

### 4.1. Numerical Example 1

Suppose that in a supplier selection decision, four alternative suppliers $\left(A_{1},A_{2},A_{3},A_{4}\right)$ are identified after a qualification process, short visits, and in-depth research. The criteria considered are: supplier performance ($C_{1}$), supply stability ($C_{2}$), and developing innovation capabilities ($C_{3}$). The criteria have different importance but the weight of the criteria is unknown. The decision information is represented by intuitionistic fuzzy numbers. 

The calculation steps based on the proposed method are as follows:

Step1: The decision information of all experts is aggregated into the intuitionistic normal cloud through the backward cloud generation algorithm for intuitionistic normal clouds (for the sake of brevity, we have omitted the detailed description of this step). The criterion values of the four alternative suppliers under the three criteria are expressed as the intuitionistic normal cloud matrix in Table 1.

Step2: Determine the positive ideal solution and negative ideal solution of each criterion. According to Equation (14), we can obtain the positive ideal solution of the decision cloud matrix as follows:$$Y_{1}^{+}=(\langle 6.41,[0.79,0.13]\rangle ,0.47,0.09)\;Y_{2}^{+}=(\langle 8.12,[0.81,0.11]\rangle ,0.88,0.29)\;Y_{3}^{+}=(\langle 6.14,[0.79,0.13]\rangle ,0.35,0.08)$$

According to Equation (15), we can obtain the negative ideal solution of the decision cloud matrix as follows:$$Y_{1}^{-}=(\langle 4.56,[0.62,0.32]\rangle ,0.81,0.23)\;Y_{2}^{-}=(\langle 6.71,[0.51,0.36]\rangle ,1.14,0.41)\;Y_{3}^{-}=(\langle 4.56,[0.53,0.28]\rangle ,0.59,0.12)$$

Step3: Determine the criterion weight. According to Equation (16), we can obtain the distance matrix between each scheme cloud and the ideal optimal cloud as follows:$$D=\left[\begin{array}{ccc} 1.4360 & 2.1353 & 1.4489\\ 0.6781 & 1.1582 & 0.5983\\ 1.5286 & 2.1003 & 1.2543\\ 1.4617 & 1.1624 & 0.7167 \end{array}\right]$$

According to Equations (17) and (18), we obtain the criterion weight as
w=(0.2821,0.2898,0.4281)

Step4: Take the compromise coefficient u as 0.6. The group utility value $S_{i}$, individual regret value $R_{i}$, and compromise value $Q_{i}$ are obtained by Equations (19)–(21) as shown in Table 2.

According to the value of $Q_{i}$, the alternative supplier is ranked as $A_{2}>A_{4}>A_{3}>A_{1}$. At the same time, according to the sorting results of $S_{i}$ and $R_{i}$, $A_{2}$ is still the optimal alternative, and $Q(A_{4})-Q(A_{2})=0.3679>1/3$. Therefore, the best supplier is $A_{2}$.

### 4.2. Comparative Analysis

#### 4.2.1. Error Analysis of Backward Cloud Generation Algorithm

The effectiveness of the backward cloud generation algorithm directly affects the accuracy of the aggregated expert evaluation information, so the error analysis of the algorithm is required. Considering that the backward cloud generation algorithm based on intuitionistic fuzzy numbers has not been found in the existing research, it cannot be analyzed by the method of similar comparison. We designed experiments based on the intuitionistic normal cloud correlation theory to analyze the proposed backward cloud generation algorithm. The experimental steps are as follows.

Step1: Generate a normal random number $En^{\prime }{}_{i}$ with En as the expectation and He as the standard deviation. Generate a normal random number $x_{i}$ with Ex as the expectation and $\left|En^{\prime }{}_{i}\right|$ as the standard deviation;

Step2: Generate a uniformly distributed random number $r_{i}$ in the interval [u,1−v];

Step3: Calculate membership degree $u_{i}=u\times e^{\left(-{\left((x_{i}-Ex)/En^{\prime }{}_{i}\right)}^{2}\right)}$ and nonmembership degree $v_{i}=1-(1-v)\times e^{\left(-{\left((x_{i}-Ex)/En^{\prime }{}_{i}\right)}^{2}\right)}$;

Step4: Output the intuitionistic fuzzy number $\left(x_{i},[u_{i},v_{i}]\right)$;

Step5: Repeat Step1–Step4 until a sufficient number of intuitionistic fuzzy numbers are generated.

In this experiment, let Ex=5, u=0.51, v=0.42, En=0.78, He=0.34; we generate the intuitionistic fuzzy numbers through the above algorithm, and then use the proposed backward cloud generation algorithm to obtain the estimated numerical features of the intuitionistic normal cloud and analyze the error of the algorithm.

(1)Algorithm validity analysis

The above method is used to generate 1000 intuitionistic fuzzy numbers as cloud droplet samples, and then the estimated numerical features of intuitionistic normal clouds are calculated by the proposed backward cloud generation algorithm. Let the number of experiments be 100, and the absolute error of each numerical feature in each experiment is shown in Figure 3.

From Figure 3, we can find that the mean absolute error and mean relative error of Ex are 0.0172 and 3.4%, respectively; the mean absolute error and mean relative error of u are 0.0256 and 5%, respectively; the mean absolute error and mean relative error of v are 0.0291 and 6.9%, respectively; the mean absolute error and mean relative error of En are 0.0157 and 2%, respectively; and the mean absolute error and mean relative error of He are 0.0211 and 6.2%, respectively. Although the error of each numerical feature fluctuates to a certain extent, high-precision numerical features of the intuitionistic normal cloud can be obtained, which shows the effectiveness of the proposed backward cloud generation algorithm.

(2)Algorithm adaptability analysis

Hyperentropy is an uncertainty measurement of entropy that can be used to describe the thickness of a cloud, and its magnitude affects the state distribution of the cloud. As hyperentropy increases, the state of discrete cloud droplets transforms from normal to pan-normal distribution and then transitions to the atomized state. To verify the adaptability of the backward cloud generation algorithm, let hyperentropy take a range of values He={0,0.02,0.05,0.08,0.1,0.15,0.2,0.25,0.3,0.4,0.6,0.8,1,1.2,1.5}, and repeat the experiment 100 times for each He. Figure 4 depicts the change in the mean absolute error of the estimates of five numerical features as hyperentropy increases.

The mean absolute error of the estimates of the five numerical features increases as He increases. However, when He≤En, the error in the estimated value increases very slowly and the value is small, which is within the acceptable range. Therefore, it can be considered that the proposed algorithm is not only suitable for cloud droplet samples with normal distribution, but also has strong adaptability for cloud droplets with pan-normal distribution and atomized state. That is, the proposed algorithm can accurately restore the numerical features of the intuitionistic normal cloud as long as the sample slightly satisfies the normal distribution.

(3)Effect of cloud droplet number on error

In the backward cloud generation algorithm, the number of cloud droplets directly affects the comprehensiveness of cloud model information. In order to analyze the effect of cloud droplet number on the error, the range of values of cloud droplet number is set to N=[10:100:3000], and each experiment is repeated 100 times. The effect of cloud droplet number on the error is shown in Figure 5.

As the number of cloud droplets increases, the errors of the estimates of the five numerical features keep decreasing and converge to 0. Meanwhile, the estimated numerical features with high accuracy can be obtained at the initial stage when the number of cloud droplets is small. This feature determines that the proposed backward cloud generation algorithm can have good application scenarios in intuitionistic fuzzy multicriteria group decision problems, thus improving the accuracy of decision making without increasing the workload.

#### 4.2.2. Sensitivity Analysis

In the multicriteria group decision problem, we need to test the robustness of the proposed decision model by sensitivity analysis. Specifically, we need to observe the impact of potential changes in criterion weights on the ranking results, which is the key to effectively utilizing the model and enabling quantitative decision making. We used the perturbation method [44] for sensitivity analysis of the criterion weights, the main idea of which is to observe the corresponding changes in the ranking results of the alternatives after a small perturbation of the criterion weights in the decision. The initial weight of the decision criterion $c_{j}$ is $w_{j}$, and the weight after the disturbance is $w_{j}^{\prime }=\zeta w_{j}$, where $0\leq w_{j}^{\prime }\leq 1$. The variation interval of the parameter ζ is $0\leq \zeta \leq 1/w_{j}$. According to the normalization of the weights, the weights of the other criteria will change accordingly due to the change of $w_{j}$, denoted as $w_{k}^{\prime }=\varphi w_{k},k\neq j$, and $w_{j}$ satisfies
(22)$$w_{j}^{\prime }+\sum_{k\neq j,k=1}^{n}w_{k}^{\prime }=1\Rightarrow \zeta w_{j}+\varphi \sum_{k\neq j,k=1}^{n}w_{k}=1$$

According to Equation (22), we can obtain $\varphi =(1-\zeta w_{j})/(1-w_{j})$. Let the perturbation range of the criterion weight be −50% to 50%; the perturbation step size is 5%, and a total of 60 perturbation experiments are carried out. The results of the sensitivity analysis are shown in Figure 6.

Obviously, the $Q_{i}$ value of supplier $A_{2}$ is the smallest in all 60 experiments. The ranking result of supplier $A_{4}$ is always 2nd. Supplier $A_{1}$ ranks 4th out of 54 experiments (ranked 3rd out of Experiments 17–20). Supplier $A_{1}$ and supplier $A_{3}$ are more sensitive to criteria $c_{1}$. In summary, the proposed decision-making method is relatively insensitive to changes in the evaluation information, and the ranking results only changed in 4 out of 60 perturbation experiments, with a change probability of 6.7%. Meanwhile, the position of the optimal supplier has remained unchanged, indicating that the model has good robustness.

#### 4.2.3. Comparative Analysis

Through a review of existing research, the Monte Carlo simulation is the main method to deal with multicriteria group decision problems in the intuitionistic normal cloud environment [27,45,46]. The basic idea of the Monte Carlo simulation is to rank the alternatives according to the score statistics of cloud drop samples. In order to verify the effectiveness of the proposed method, the Monte Carlo simulation method in [27] is adopted to solve this problem. Repeat the Monte Carlo simulation experiment 10 times, where each experiment produces 5000 cloud droplets. The average value $G_{i}^{\left(mean\right)}$ of cloud droplet score $G_{i}$ in 10 experiments is used as the final decision result. In addition, grey relational analysis (GRA) and technique for order preference by similarity to an ideal solution (TOPSIS) are extended to perform experiments in the intuitionistic normal cloud environment to achieve the comparison of ranking methods. When the GRA method is applied, the positive ideal solution is taken from the intuitionistic normal cloud matrix as the reference sequence, and the correlation $\xi_{i}$ between each alternative and the reference sequence is calculated based on the proposed cloud distance measure, where larger values of $\xi_{i}$ are associated with better alternatives $A_{i}$. When the TOPSIS method is applied, the cloud distances ($D_{i}^{+}$ and $D_{i}^{-}$) of each alternative from the positive and negative ideal solutions are calculated, respectively. The ranking results are obtained from the relative proximity distances $D_{i}$ of each alternative to the negative ideal solution in descending order. The parameters of all algorithms used in the research are shown in Table 3. The ranking results of the different methods are summarized in Table 4. The comparison of the different methods is shown in Figure 7.

From Table 4 and Figure 7, we can find that the optimal supplier obtained by the four methods is $A_{2}>A_{4}>A_{3}>A_{1}$. The ranking results of the Monte Carlo simulation and GRA are $A_{2}$, which are consistent with the results of the proposed method. The effectiveness of the proposed method can be verified. However, it can be found that the ranking results of the Monte Carlo simulation are unstable in 10 experiments. The results of 3 out of 10 experiments (Experiment 4, 7, 10) are $A_{2}>A_{4}>A_{1}>A_{3}$, and the results of 7 out of 10 experiments (Experiment 1, 2, 3, 5, 6, 8, 9) are $A_{2}>A_{4}>A_{3}>A_{1}$. The main reason is that the difference between $A_{1}$ and $A_{3}$ is not obvious and the cloud droplet scores are relatively close, while the randomness of the cloud droplet samples in the Monte Carlo simulation generates a certain degree of error, making it difficult to compare similar alternatives. In addition, the Monte Carlo simulation consumes more resources and time. The ranking results obtained by TOPSIS differ from the proposed method. The main reason is that the results obtained by TOPSIS are comprehensively determined by the distances between the alternatives and the positive and negative ideal solutions, so the ranking results may not be the closest to the ideal solutions. Although the ranking results obtained by GRA are consistent with the proposed method, they only consider the curve similarity between the alternative and the reference sequence, and the decision results may change depending on the selection of different reference sequences and correlation coefficients. In summary, the proposed method has outstanding features in processing intuitionistic normal cloud decision information. It has obvious advantages over the existing methods in terms of feasibility, stability, and effectiveness, and the obtained decision results are more scientific and reasonable.

### 4.3. Numerical Example 2

To further verify the superiority and limitations of the proposed method and to avoid the coincidence brought by a single numerical example, we added a bigger general example to increase the convincing power. This numerical example contains eight alternatives $\left(A_{1},A_{2},A_{3},A_{4},A_{5},A_{6},A_{7},A_{8}\right)$ and six criteria $\left(C_{1},C_{2},C_{3},C_{4},C_{5},C_{6}\right)$ with unknown criteria weight and with different importance. The intuitionistic normal cloud decision matrix obtained after the processing of the proposed backward cloud generation algorithm is shown in Table 5.

According to the method of determining the criterion weight based on cloud distance entropy, we can obtain the criterion weight as
w=(0.1594,0.1739,0.2530,0.2613,0.0545,0.0978)

The ranking results obtained according to INC-GRA, INC-TOPSIS, INC-Monte Carlo, and the proposed method are shown in Table 6, where the parameters of all algorithms are consistent with Table 3.

From Table 6, we can find that the optimal alternatives obtained by INC-GRA, INC- Monte Carlo, and the proposed method are all $A_{6}$, and the worst alternatives are all $A_{3}$, and the ranking results are basically the same, which further verifies the effectiveness of the proposed method.

In addition, the coefficient of ranking similarity is a useful way to compare the ranking of different methods in decision-making problems [47]. To compare the differences between the different rankings in depth, we used the results obtained from Monte Carlo simulation as the reference ranking and measured the ranking similarity between the different method rankings and the reference ranking using the Spearman’s coefficient $r_{s}$, the weighted rank measure of correlation $r_{w}$ and the WS coefficient of ranking similarity WS, respectively [48]. The obtained results are summarized in Table 7.
(23)$$r_{s}=1-\frac{6\cdot \sum {\left(R_{xi}-R_{yi}\right)}^{2}}{n\cdot (n^{2}-1)}$$
(24)$$r_{w}=1-\frac{6\cdot \sum_{i=1}^{n}{\left(R_{xi}-R_{yi}\right)}^{2}((n-R_{xi}+1)+(n-R_{yi}+1))}{n^{4}+n^{3}-n^{2}-n}$$
(25)$$WS=1-\sum_{i=1}^{n}\left(2^{-R_{xi}}\cdot \frac{\left|R_{xi}-R_{yi}\right|}{\max\left\{\left|1-R_{xi}\right|,\left|N-R_{xi}\right|\right\}}\right)$$

The all measures show the same relationship between rankings, i.e., Spearman’s coefficient $r_{s}$, the weighted rank measure of correlation $r_{w}$ and the WS coefficient of ranking similarity WS are the highest for INC-GRA and the proposed method, and the worse for INC-TOPSIS. This comparison proves that the rankings obtained by GRA and the proposed method are better than those obtained by TOPSIS.

Moreover, VIKOR can take into account both group utility and individual regret. Its significant advantage over other methods is its ability to reflect the decision maker’s preferences through different compromise coefficients in decision making, which improves the flexibility and elasticity of decision making. When the compromise coefficient is small, it indicates that the decision maker prefers individual regret value in decision making and vice versa for group utility value. To analyze the influence of decision preferences on the assessment results, the results obtained by selecting different compromise coefficients are shown in Figure 8.

As shown in Figure 8, as the compromise coefficient increases, the decision maker’s preferences gradually transform from individual regret to group utility, resulting in increasing rankings for $A_{1}$ and $A_{8}$, while decreasing rankings for $A_{2}$ and $A_{5}$. This result indicates that the proposed method can reflect the different preferences of decision makers and has better flexibility and elasticity in decision making. At the same time, the best alternative is always $A_{6}$ and the worst alternative is always $A_{3}$ for different compromise coefficients, which indicates that the method is able to maintain good stability while having a certain degree of flexibility. Compared to VIKOR, other modern methods make it difficult to adjust the results accordingly to the decision maker’s preferences, ignoring the interactivity and flexibility that exist in the decision-making process.

## 5. Conclusions

In this study, a multicriteria group decision-making method based on intuitionistic normal cloud and cloud distance entropy is proposed to solve the group decision-making problem with unknown criterion weights in an intuitionistic fuzzy environment, and the effectiveness of the proposed model is verified by two numerical arithmetic examples and comparative analysis. The main contributions of the proposed model are as follows: (1) The proposed backward cloud generation algorithm deals with intuitionistic fuzzy decision-making information based on a comprehensive consideration of the fuzziness and randomness of the information. This feature facilitates the solution of group decision-making problems of different scales without loss and distortion of information. (2) The proposed concept of cloud distance entropy combines the advantages of cloud distance measure in dealing with group consistency measurement and entropy in describing information quality, which extends the application of the traditional entropy method in the field of decision making in the cloud environment. This advantage effectively solves the problems of incomplete information and insufficient objectivity in group decision making in the cloud environment. (3) The proposed intuitionistic normal cloud distance measurement is general and can accurately describe the distances and differences between cloud models, which has good practicality in the field of uncertainty decision making. (4) The model integrates group utility and individual regret, and the resulting decision results have higher stability than existing methods while maintaining validity. 

Future research will focus on the following aspects: (1) It would be very interesting to develop appropriate group decision-making models for solving evaluation problems in other domains. (2) Differences in the capabilities of different experts can be considered in the process of constructing the cloud. (3) Other intuitionistic normal cloud weighted aggregation operators such as the weighted geometric mean operator can be developed to accommodate decision-making problems in different environments. (4) Integrating the psychological behavior of decision makers into the decision-making process is also worth exploring.

## Figures and Tables

**Figure 1 entropy-24-01396-f001:**
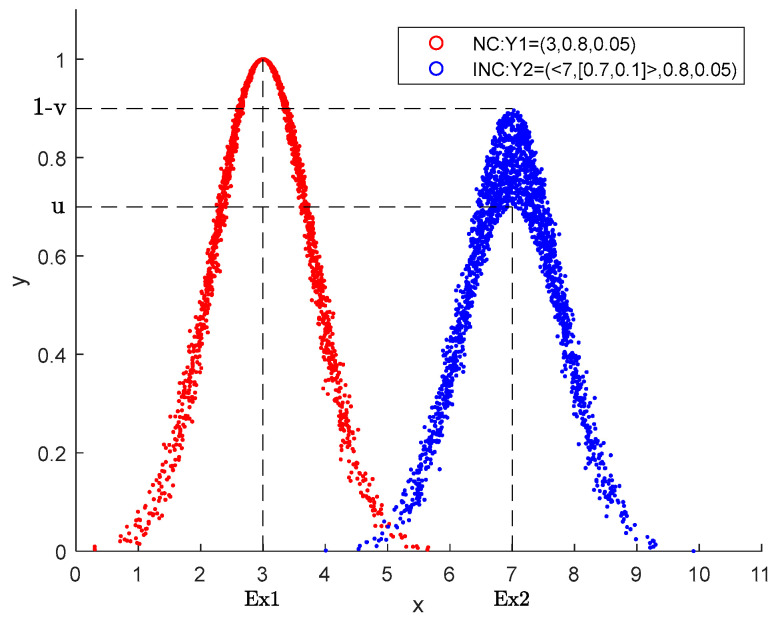
Comparison of $Y_{1}$ and $Y_{2}$.

**Figure 2 entropy-24-01396-f002:**
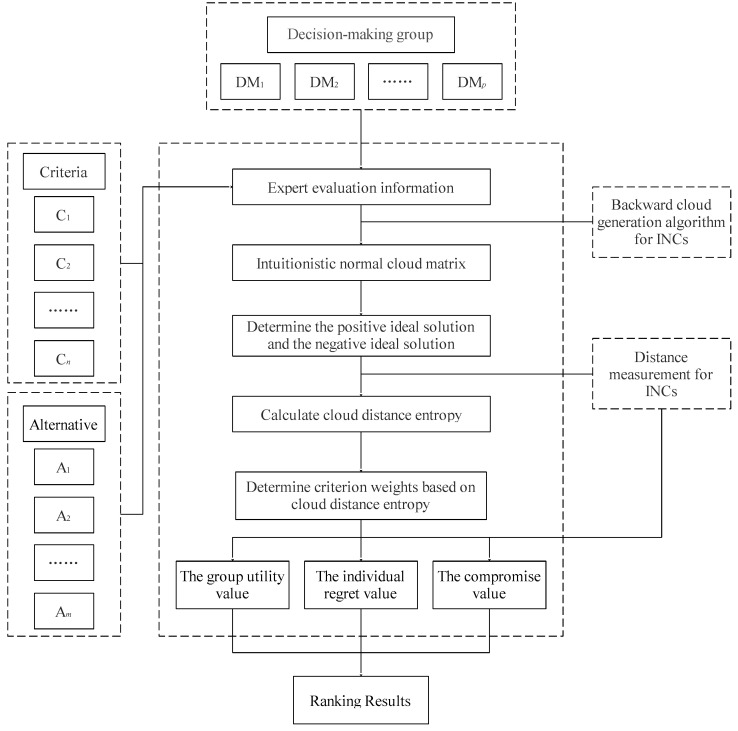
The flowchart diagram of the proposed method.

**Figure 3 entropy-24-01396-f003:**
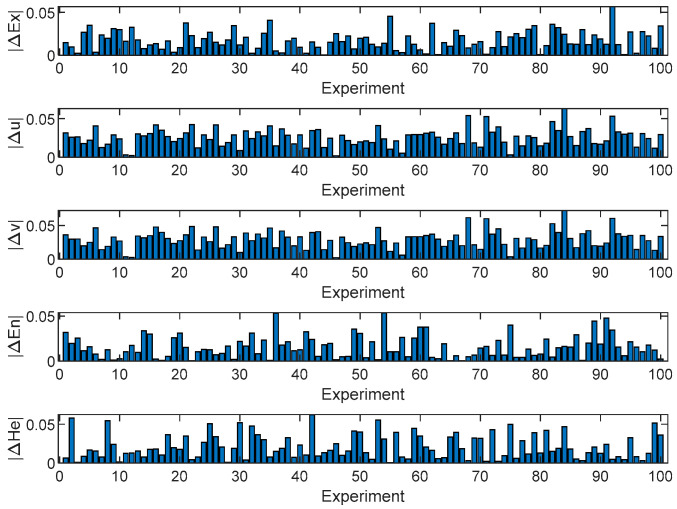
The absolute error of cloud numerical features.

**Figure 4 entropy-24-01396-f004:**
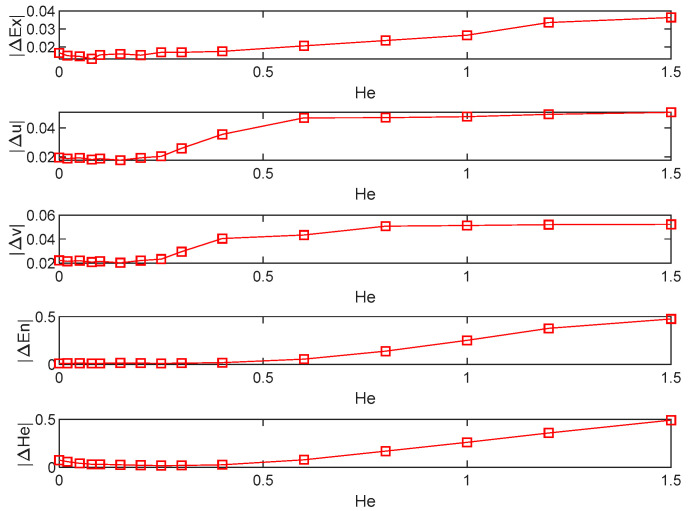
Changes in mean absolute error as hyperentropy increases.

**Figure 5 entropy-24-01396-f005:**
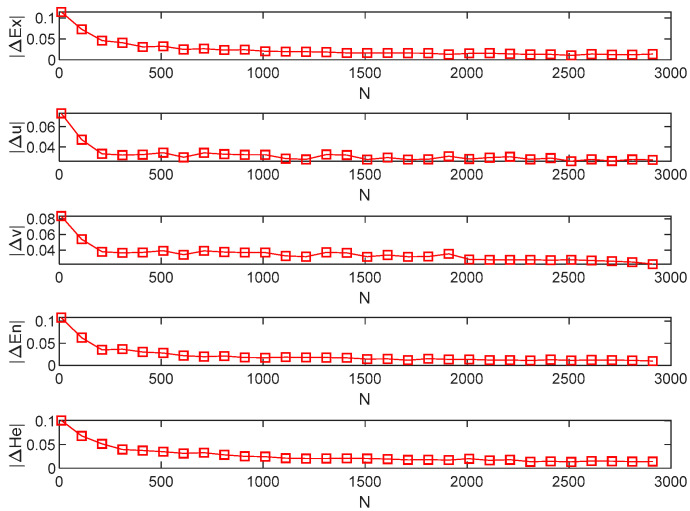
Changes in error as the number of cloud droplets increases.

**Figure 6 entropy-24-01396-f006:**
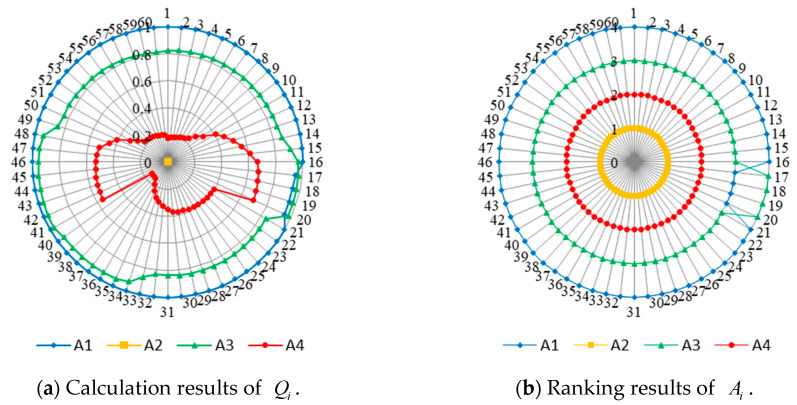
Sensitivity analysis results.

**Figure 7 entropy-24-01396-f007:**
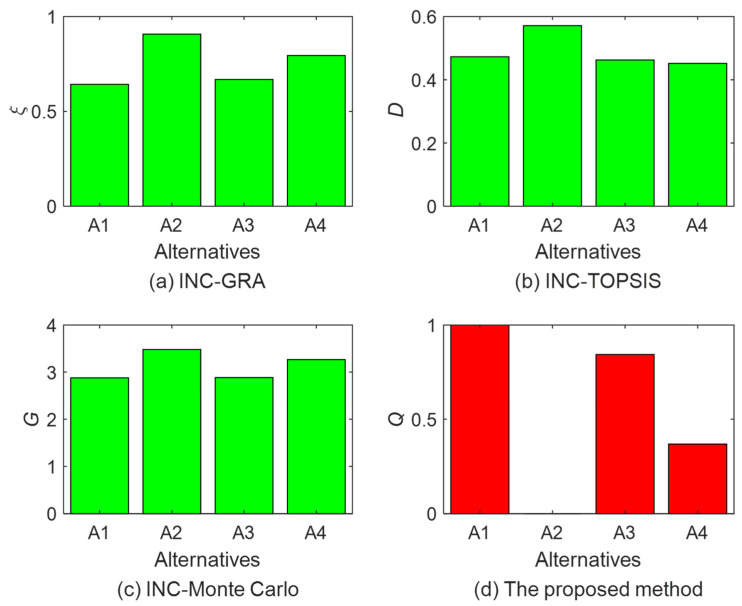
Comparison of different methods.

**Figure 8 entropy-24-01396-f008:**
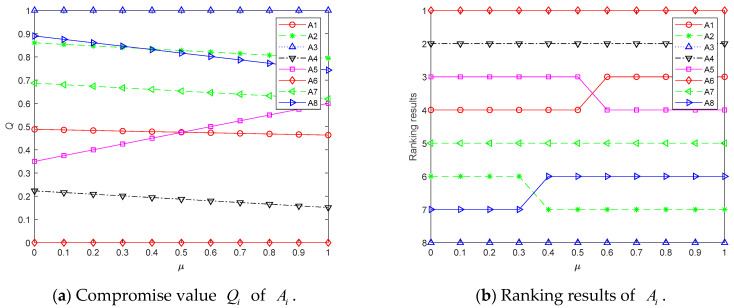
Comparison of results for different compromise coefficients.

**Table 1 entropy-24-01396-t001:** Intuitionistic normal cloud decision matrix.

	$C_{1}$	$C_{2}$	$C_{3}$
$A_{1}$	(<5.03, [0.71, 0.14]>, 0.78, 0.2)	(<7.35, [0.52, 0.21]>, 1.14, 0.33)	(<4.56, [0.74, 0.14]>, 0.46, 0.08)
$A_{2}$	(<6.41, [0.67, 0.23]>, 0.47, 0.23)	(<7.38, [0.71, 0.13]>, 0.93, 0.41)	(<5.44, [0.79, 0.13]>, 0.55, 0.12)
$A_{3}$	(<4.56, [0.79, 0.13]>, 0.81, 0.19)	(<8.12, [0.51, 0.36]>, 1.06, 0.29)	(<6.14, [0.53, 0.28]>, 0.59, 0.11)
$A_{4}$	(<5.79, [0.62, 0.32]>, 0.66, 0.09)	(<6.71, [0.81, 0.11]>, 0.88, 0.38)	(<5.86, [0.68, 0.17]>, 0.35, 0.09)

**Table 2 entropy-24-01396-t002:** Decision results of the proposed method.

	$S_{i}$	$R_{i}$	$Q_{i}$
$A_{1}$	0.6739	0.2819	1.0000
$A_{2}$	0.3165	0.1164	0.0000
$A_{3}$	0.6441	0.2441	0.8585
$A_{4}$	0.4376	0.1821	0.3621

**Table 3 entropy-24-01396-t003:** Parameters of all algorithms used in the research.

Algorithm	Parameters	
GRA	Grey correlation coefficient δ	0.5
Monte Carlo simulation	Number of cloud droplets N	5000
	Number of experiments E	10
VIKOR	Compromise coefficient u	0.6

**Table 4 entropy-24-01396-t004:** Ranking results obtained by different methods.

Method		$A_{1}$	$A_{2}$	$A_{3}$	$A_{4}$	Ranking Result
INC-GRA	$\xi_{i}$	0.6428	0.9069	0.6686	0.7947	$A_{2}>A_{4}>A_{3}>A_{1}$
INC-TOPSIS	$D_{i}^{+}$	1.6440	0.7830	1.5767	1.0559	$A_{2}>A_{1}>A_{3}>A_{4}$
	$D_{i}^{-}$	1.4749	1.0426	1.3565	0.8703	
	$D_{i}$	0.4729	0.5711	0.4625	0.4518	
INC-Monte Carlo in [27]	$G_{i}^{\left(1\right)}$	2.8729	3.5170	2.8854	3.2574	$A_{2}>A_{4}>A_{3}>A_{1}$
	$G_{i}^{\left(2\right)}$	2.8898	3.4545	2.8960	3.2764	
	$G_{i}^{\left(3\right)}$	2.8521	3.4825	2.8611	3.2863	
	$G_{i}^{\left(4\right)}$	2.9062	3.4863	2.8659	3.2559	
	$G_{i}^{\left(5\right)}$	2.8685	3.4706	2.9048	3.2588	
	$G_{i}^{\left(6\right)}$	2.8785	3.4514	2.8924	3.2759	
	$G_{i}^{\left(7\right)}$	2.9033	3.4760	2.8589	3.2792	
	$G_{i}^{\left(8\right)}$	2.8625	3.4758	2.8887	3.2648	
	$G_{i}^{\left(9\right)}$	2.8712	3.5078	2.8914	3.2448	
	$G_{i}^{\left(10\right)}$	2.8836	3.4858	2.8649	3.2605	
	$G_{i}^{\left(mean\right)}$	2.8789	3.4808	2.8849	3.2660	
The proposed method	$S_{i}$	0.6739	0.3165	0.6441	0.4376	$A_{2}>A_{4}>A_{3}>A_{1}$
	$R_{i}$	0.2819	0.1164	0.2441	0.1821	
	$Q_{i}$	1.0000	0.0000	0.8585	0.3621	

**Table 5 entropy-24-01396-t005:** Intuitionistic normal cloud decision matrix.

	$C_{1}$	$C_{2}$	$C_{3}$
$A_{1}$	(<5.84, [0.58, 0.32]>, 2.24, 0.39)	(<6.47, [0.65, 0.25]>, 2.35, 0.38)	(<4.28, [0.59, 0.24]>, 2.17, 0.39)
$A_{2}$	(<4.91, [0.51, 0.39]>, 2.08, 0.43)	(<6.53, [0.63, 0.27]>, 2.28, 0.37)	(<5.33, [0.53, 0.37]>, 2.02, 0.45)
$A_{3}$	(<3.83, [0.6, 0.3]>, 2.6, 0.29)	(<3.84, [0.64, 0.35]>, 2.26, 0.39)	(<3.25, [0.51, 0.43]>, 2.66, 0.23)
$A_{4}$	(<7.95, [0.72, 0.18]>, 2.59, 0.28)	(<5.25, [0.6, 0.25]>, 2.66, 0.23)	(<5.56, [0.76, 0.21]>, 2.35, 0.37)
$A_{5}$	(<4.19, [0.51, 0.38]>,2.39, 0.33)	(<4.3, [0.75, 0.1]>, 2.96, 0.13)	(<5.38, [0.58, 0.37]>, 2.41, 0.29)
$A_{6}$	(<4.95, [0.8, 0.1]>, 2.96, 0.13)	(<8.65, [0.75, 0.15]>, 2.69, 0.27)	(<6.14, [0.81, 0.12]>, 2.44, 0.16)
$A_{7}$	(<4.08, [0.7, 0.12]>, 2.37, 0.25)	(<5.14, [0.68, 0.21]>, 2.15, 0.34)	(<4.76, [0.73, 0.15]>, 2.16, 0.33)
$A_{8}$	(<6.33, [0.68, 0.23]>, 2.14, 0.31)	(<7.14, [0.55, 0.34]>, 2.41, 0.33)	(<3.98, [0.61, 0.35]>, 2.31, 0.27)
	$C_{4}$	$C_{5}$	$C_{6}$
$A_{1}$	(<4.97, [0.72, 0.23]>, 1.97, 0.46)	(<6.19, [0.68, 0.22]>, 2.28, 0.51)	(<7.68, [0.71, 0.22]>, 2.65, 0.34)
$A_{2}$	(<4.29, [0.56, 0.41]>, 2.09, 0.43)	(<4.28, [0.63, 0.27]>, 2.69, 0.48)	(<5.82, [0.59, 0.31]>, 1.99, 0.45)
$A_{3}$	(<4.26, [0.6, 0.31]>, 2.46, 0.23)	(<4.56, [0.81, 0.12]>, 1.98, 0.39)	(<6.17, [0.72, 0.23]>, 1.94, 0.47)
$A_{4}$	(<6.43, [0.7, 0.27]>, 2.16, 0.37)	(<4.97, [0.72, 0.18]>, 2.34, 0.42)	(<6.31, [0.68, 0.21]>, 2.38, 0.29)
$A_{5}$	(<5.18, [0.67, 0.12]>, 2.36, 0.33)	(<5.41, [0.66, 0.3]>, 2.14, 0.46)	(<7.04, [0.66, 0.32]>, 2.09, 0.36)
$A_{6}$	(<6.07, [0.8, 0.13]>, 2.28, 0.27)	(<6.74, [0.58, 0.34]>, 2.55, 0.37)	(<6.84, [0.58, 0.39]>, 2.46, 0.41)
$A_{7}$	(<3.84, [0.74, 0.21]>, 2.66, 0.49)	(<4.97, [0.62, 0.21]>, 2.61, 0.53)	(<6.36, [0.73, 0.21]>, 2.17, 0.32)
$A_{8}$	(<4.26, [0.55, 0.42]>, 2.04, 0.28)	(<5.31, [0.59, 0.24]>, 2.48, 0.32)	(<5.38, [0.67, 0.13]>, 2.27, 0.44)

**Table 6 entropy-24-01396-t006:** Ranking results obtained by different methods.

Method	INC-GRA	INC-TOPSIS	INC-Monte Carlo	The Proposed Method
	$\xi_{i}$	$D_{i}$	$G_{i}^{\left(mean\right)}$	$Q_{i}$
$A_{1}$	0.6347	0.4754	2.7391	0.4732
$A_{2}$	0.5452	0.3836	2.2320	0.8209
$A_{3}$	0.5021	0.3940	1.8725	1.0000
$A_{4}$	0.7774	0.6288	3.2142	0.1801
$A_{5}$	0.5988	0.4355	2.4931	0.4999
$A_{6}$	0.8650	0.6016	3.6403	0.0000
$A_{7}$	0.5926	0.3920	2.5239	0.6463
$A_{8}$	0.5586	0.4442	2.3559	0.8019
Ranking result	$\begin{array}{l} A_{6}>A_{4}>A_{1}>A_{5}\\ >A_{7}>A_{8}>A_{2}>A_{3} \end{array}$	$\begin{array}{l} A_{4}>A_{6}>A_{1}>A_{8}\\ >A_{5}>A_{3}>A_{7}>A_{2} \end{array}$	$\begin{array}{l} A_{6}>A_{4}>A_{1}>A_{7}\\ >A_{5}>A_{8}>A_{2}>A_{3} \end{array}$	$\begin{array}{l} A_{6}>A_{4}>A_{1}>A_{5}\\ >A_{7}>A_{8}>A_{2}>A_{3} \end{array}$

**Table 7 entropy-24-01396-t007:** Comparison of rank correlation measurement (in respect of reference ranking).

Method	Measure of Rank Correlation		
INC-GRA	$r_{s}$	$r_{w}$	WS
	0.9762	0.9762	0.9766
INC-TOPSIS	0.7619	0.8095	0.8314
The proposed method	0.9762	0.9762	0.9766

## Data Availability

Not applicable.

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
