# Peer review of "Multicriteria Group Decision Making Based on Intuitionistic Normal Cloud and Cloud Distance Entropy"

_entropy, 2022, doi:10.3390/e24101396_

Round 1

Reviewer 1 Report

i am grateful to have the opportunity to review your manuscript.

i am pleased to share that i am satisfied with it as i believe you have cogently argued your case and have taken care to structure and signpost the critical inflexion points within your manuscript.

I found the manuscript systematically organised, with clear signposts to help the reader follow the arguments and reasoning of the authors.

The rationale for the study is well situated within existing literature, with the gaps in literature clearly presented.

Finally, the use of the numerical example and subsequent comparative analysis helps to ground the abstract theorem in to concrete utility, which further strengthens the merit of the manuscript.

Reviewer 2 Report

In this manuscript, the authors proposed a multi-criteria group decision-making method based on intuitionistic normal cloud and cloud distance entropy. Further, they introduced the distance measurement of the cloud model into the information entropy theory.  Finally, they explored a numerical example to show the effectiveness and practicality of the proposed method. I have some comments to improve the quality of the manuscript.

1.  Please use the full forms of the terms, including VIKOR, TOPSIS, etc.

2. There are several grammatical mistakes in the manuscript which should be corrected with the help of a native English speaker, for example:

 On page 1, line 33, please replace “the Vague set” by “the vague set”.

 On page 2, line 50, please replace “The Cloud” by “The cloud”.

3.      Add motivations and contributions of your work as separate subsections in the Introduction section.

4.      Add a flowchart diagram of your proposed method.

5.      For better demonstration, please add graphical representation of your discussed comparison results.

6.      Please clearly describe 3-5 future directions of your work at the end of conclusion section.

7.      Literature review is weak. Please read and add latest references regarding group decision-making like

Group Generalized q-Rung Orthopair Fuzzy Soft Sets: New Aggregation Operators and Their Applications, Mathematical Problems in Engineering, 2021(2021), Article ID 5672097; Group decision-making with Fermatean fuzzy soft expert knowledge, Artificial Intelligence Review, 55(2022), 5349–5389; Hybrid group decision-making technique under spherical fuzzy N-soft expert sets, Artificial Intelligence Review 55(5)(2022), 4117-4163; New aggregation operators on group-based generalized intuitionistic fuzzy soft sets, Soft Computing, 25(2021), 13353–13364.

Reviewer 3 Report

The two most common types of uncertainty are randomness and fuzziness. Therefore, the authors propose multi-criteria group decision-making (MCGDM) method based on intuitionistic normal cloud and cloud distance entropy. The proposal is as follows:

- the backward cloud generation algorithm for intuitionistic normal clouds is proposed to transform the intuitionistic fuzzy decision information given by all experts into an intuitionistic normal cloud matrix to avoid the loss and distortion of information;

- the distance measurement of the cloud model is introduced into the information entropy theory, and the concept of cloud distance entropy is proposed. 

- then the distance measurement for intuitionistic normal clouds based on numerical features is defined, and its properties are discussed, based on which the criterion weight determination method under intuitionistic normal cloud information is proposed. 

- in addition, the VIKOR method, which integrates group utility and individual regret, is extended to the intuitionistic normal cloud environment, and thus the ranking results of the alternatives are obtained. 

- the effectiveness and practicality of the proposed method are demonstrated by a numerical example.

The paper is well-written scientific work. However, some aspects need to be improved before accepting. The first of all is better explain the effectiveness of the proposed approach. The presented example is not enough. Therefore I propose to add a bigger example to illustrate this method's superiority and limitations.

Moreover, The rankings is better to compare by using a new coefficient of rankings similarity in decision-making problems, it can be an easy find, e.g., 10.1007/978-3-030-50417-5_47. Of course, r_w coefficient is also nice to see for deeper analysis. The second question is: why do you select the VIKOR method? What is the superiority and limitations of this selection? Why do you not use some more modern methods like SPOITIS, COMET, CODAS-COMET or TOPSIS-DARIA? In my opinion, it should be better explained (perhaps it should be a separate part about selection. The last comment is about future research directions. Please extend it.

Reviewer 4 Report

This paper develops an entropy method based on intuitionistic normal cloud and cloud distance entropyan algorithm for multi-criteria group decision-making It belongs to a fuzzy distance entropy+VIKOR group computing compared to other crisp value approaches by the use of weighted averaging function for group decision making to attain some advantages on fuzzy semantics. The work is well written and well presented, the idea of working with case studies is interesting. It also presents some preliminary experimental results showing the proposed method leads to much elastic way to support the group decision making process. 

I have the following concerns about the paper. 

1.You need to add one Table to show all algorithm's parameters that used in this research. 

2.The literature has to be  updated with some recent papers focused on the fields dealt with the manuscript. Foe example,

Garg, H., Kumar, K. Some Aggregation Operators for Linguistic Intuitionistic Fuzzy Set and its Application to Group Decision-Making Process Using the Set Pair Analysis. Arab J Sci Eng 43, 3213–3227 (2018).

H. Garg and K. Kumar, "Linguistic Interval-Valued Atanassov Intuitionistic Fuzzy Sets and Their Applications to Group Decision Making Problems," in IEEE Transactions on Fuzzy Systems, vol. 27, no. 12, pp. 2302-2311, Dec. 2019

Round 2

Reviewer 2 Report

Please correct the following minor mistakes:

1. On page 1, line 28,  please correct typos "Man1y scholars".

2. On page 3, lines 112-113, 109-110 & line 122, please use abbreviation "MCGDM" instead of "multi-criteria group decision-making",

3. From lines 111-123, Please use the abbreviation "NC" instead of "normal cloud".

Reviewer 3 Report

The paper has been improved.